# Alternative Lengthening of Telomeres: Lessons to Be Learned from Telomeric DNA Double-Strand Break Repair

**DOI:** 10.3390/genes12111734

**Published:** 2021-10-29

**Authors:** Thomas Kent, David Clynes

**Affiliations:** 1Molecular Haematology Unit, Radcliffe Department of Medicine, The MRC Weatherall Institute of Molecular Medicine, John Radcliffe Hospital, University of Oxford, Oxford OX3 9DS, UK; thomas.kent@rdm.ox.ac.uk; 2Department of Oncology, The MRC Weatherall Institute of Molecular Medicine, John Radcliffe Hospital, University of Oxford, Oxford OX3 9DS, UK

**Keywords:** telomeres, alternative lengthening of telomeres, ATRX, break-induced replication, double-strand break repair

## Abstract

The study of the molecular pathways underlying cancer has given us important insights into how breaks in our DNA are repaired and the dire consequences that can occur when these processes are perturbed. Extensive research over the past 20 years has shown that the key molecular event underpinning a subset of cancers involves the deregulated repair of DNA double-strand breaks (DSBs) at telomeres, which in turn leads to telomere lengthening and the potential for replicative immortality. Here we discuss, in-depth, recent major breakthroughs in our understanding of the mechanisms underpinning this pathway known as the alternative lengthening of telomeres (ALT). We explore how this gives us important insights into how DSB repair at telomeres is regulated, with relevance to the cell-cycle-dependent regulation of repair, repair of stalled replication forks and the spatial regulation of DSB repair.

## 1. Aberrant Telomeric BIR—A Potential Route to Cancer

The formation of DSBs, whether emanating from chromosome breakage, replication fork collapse or telomere deprotection, constitutes a major threat to the stability of the genome, which if not faithfully repaired can have profound consequences. Indeed, loss of key factors in the DNA double-strand break repair (DSBR) response have been implicated in numerous cancers, notable examples including the breast cancer susceptibility proteins 1 and 2 (BRCA1 and BRCA2), with hereditary mutations leading to an increased predisposition to breast and ovarian cancers and to a lesser extent other cancers [1]. Extensive research has shown that there are two dominant pathways that facilitate the repair of DNA DSBs, namely Non-Homologous End Joining (NHEJ) and Homologous Recombination (HR), in addition to what have traditionally been considered backup pathways, including Microhomology-Mediated End Joining (MMEJ) and Break Induced Replication (BIR). A series of intricate control measures exist in the cell to ensure DSB repair is conducted in a timely manner and carefully coordinated depending on the stage of the cell cycle. Despite these careful checks, the process of DSB repair can itself be dangerous, leading to loss of genetic information, chromosomal translocations and, as is discussed in detail in this review, the maintenance of telomere length in the so-called ALT cancers.

The stalling and collapse of DNA replication forks during DNA replication, resulting in the generation of a DSB, represents a unique substrate for the DNA repair machinery owing to the DSB being one ended. Recent research has implicated the BIR pathway in the repair of these lesions. In line with this notion, DSBs resulting from the induction of profound replication stress through cyclin E overexpression are repaired in large part through BIR and yield gross chromosomal rearrangements [2,3]. Analogous to the dominant HR repair pathway, BIR is initiated through 5′–3′ resection of the DSB end which then invades into a homologous template and initiates DNA synthesis that can copy extensive tracts of DNA (>100 kb) until the terminus of the chromosome. Despite this the mechanism of BIR is clearly distinct from S-phase replication, with leading and lagging strands synthesized in an asynchronous manner, leading to the accumulation of long ssDNA regions. Moreover, BIR has been shown to proceed via an unusual bubble-like replication fork which results in conservative as opposed to the semi-conservative mode of DNA replication associated with S-phase replication [4]. BIR-like synthesis resulting from replication fork collapse has been documented in human cells during mitosis (Mitotic DNA synthesis (MiDAS)) [5,6] and originates from specific sites in the genome that are known to be prone to breakage under conditions of replicative stress, denoted as Common Fragile Sites (CFSs). CFSs appear to have shared properties, which likely account for why they cause problems to the replication fork under conditions of stress. This includes a propensity to adopt non-canonical DNA secondary structures, including the G-quadruplex (G4) conformation; tetrad structures formed through self-association of guanine residues in G-rich sequences (for recent review, see Reference [7]) and R-loops; and three-stranded nucleic acid structures consisting of an RNA:DNA hybrid and a displaced piece of single-stranded DNA [8]. CFSs also characteristically have a paucity of replication origins and are often late replicating, limiting the opportunity for stalled replication forks to be rescued by converging forks (for extensive review on CFSs, see Reference [9]). Telomeres, nucleoprotein structures that act as buffer sequence to prevent degradation of coding DNA as a result of the end replication problem, as well as a barrier to the recognition of DNA ends by DNA repair machinery, share many of these features. Telomeres range from 3 to 12 kb in humans and consist of guanine-rich TTAGGG_n_ repeats interspersed with a group of telomere proteins called the Shelterin complex (TRF1, TRF2, POT1, TIN1, TPP1 and RAP1) [10]. Indeed, telomeres have been found to resemble aphidicolin-induced CFSs [11], and recent work has demonstrated that the process of BIR itself promotes the formation of fragile telomeres in BLM-deficient cells and upon the formation of targeted DSBs at telomeres [12]. As such, it appears that the repair of one ended DSBs at telomeres via BIR is a somewhat double-edged sword, with the process of repair itself being potentially dangerous. The importance of carefully regulating BIR at telomeres has been exemplified by the finding that a subset of cancer cells, the so-called ALT cancers, rely on a BIR-like mechanism to maintain their telomere length [13]. This pathway is explored in depth for the remainder of this review.

## 2. The Alternative Lengthening of Telomeres Pathway

The limitless proliferative potential seen in cancers is underpinned by mutations that lead to replicative immortality [14]. The progressive shortening of telomeres by 70–100 bp per round of cellular division as a consequence of the end replication problem, wherein DNA polymerases fail to replicate the distal ends of chromosomes during cellular division, threatens long-term survival and genomic stability in cells. This shortening acts as a cellular clock that prevents the accumulation of a large number of acquired mutations and eventually leads to a state of critically short telomeres and replicative senescence that is termed the Hayflick limit [15,16].

This process of cellular proliferation regulation, however, is at odds with the requirement in complex organisms for the maintenance of long-lived cells, such as stem and early progenitor cells. In the vast majority of these cells, a system of telomere maintenance has evolved that utilizes an enzyme capable of elongating telomeres named telomerase. Telomerase, a combination of two components, the 1,132 amino acid telomerase reverse transcriptase (TERC) and an associated telomerase RNA molecule (TERT), works in a coordinated fashion to progressively elongate telomeres, counteracting the end replication problem [17,18]. Naturally, as all cells possess the capability to express telomerase and achieve replicative immortality, telomerase activity in normal cells is tightly regulated to prevent escape from senescence [19].

As many as 85% of cancers are thought to utilize this natural telomere extension mechanism to maintain their own telomeres, allowing them to evade telomere crisis [20]. A minority of cancers, the so-called ALT cancers, have established telomerase-independent mechanisms of telomere elongation [21]. It is proposed that ALT may resemble, or in fact be, the earliest form of telomere maintenance mechanism (TMM), which preceded the evolution of telomerase-dependent telomere maintenance. Indeed, it is additionally plausible that, at very low levels, ALT-like activity occurs in all cells. Nevertheless, ALT activity to the levels seen in ALT cancers has not been observed in normal functioning cells [22].

## 3. BIR at Telomeres Underpins the ALT Pathway

The ALT pathway was initially described in mutant budding yeast lacking telomerase, wherein there were two surviving populations, which were described as “types” [23]. Type I survivors display characteristic duplication of subtelomeric regions, named Y’ elements, but little elongation of telomeric sequences. In contrast, type II survivors stabilize telomere ends by elongation of telomeric sequences [24,25,26,27,28]. Historically, type I survivors were thought to be both Rad51- and Rad52-dependent, and, conversely, type II survivors were thought to be Rad52-dependent but Rad51-independent [27]. Recently published work, however, has challenged this model and shown that the majority of ALT-like survival in yeast instead relies on a unified pathway consisting of both pathways proceeding in two sequential steps. The first step entails a RAD51-dependent generation of ALT-precursors, which then subsequently undergo a second maturation phase involving RAD52-dependent telomere elongation [29].

Recent work has suggested that the mechanism and players involved in ALT telomere maintenance in human cancer cells is highly dependent on cell-cycle phase and type of telomeric lesion. This is likely a reflection of the regulation of DSB repair throughout the cell cycle, with canonical HR and NHEJ repressed during mitosis, but alternative repair pathways, such as RAD52-dependent MiDAS, permitted. Indeed, ALT in human cancer cells appears to comprise two major HR-dependent pathways which can also be broadly classified as RAD51- or RAD52-dependent. Analogous to what has been observed in yeast, telomeric BIR in ALT cells was found to be RAD51-independent and restricted to G2 and M cell cycle phases [13]. This pathway was subsequently demonstrated to be RAD52-dependent and involves the non-canonical replisome PCNA-RFC, which in turn mediates loading of Polδ to perform conservative replication of both leading and lagging strands, ultimately leading to telomere extension [13,30,31]. This conservative DNA replication is promoted by the BTR complex (BLM, TOP3A and RMI1/2) which promotes telomere dissolution, hence facilitating long tract telomere synthesis and antagonized by the SMX complex (SLX4, SLX1, MUS81-EME1, XPF-ERCC1), which conversely promotes telomere resolution [32,33]. Of note, the replication independent induction of a telomeric DSB in mitosis using a telomere targeted endonuclease can also stimulate telomere synthesis in a process coined “Break Induced Telomere Synthesis”, or BITS [13,34]. Importantly, RAD52 was found to be dispensable for BITS, suggesting that alternative telomere recombination mechanisms can compensate for RAD52 loss in ALT and that the predominant role of RAD52 in ALT telomere maintenance likely occurs downstream of replicative stress and is likely linked to the role of RAD52 in the resolution of stalled replication forks [34]. In contrast, the RAD51-dependent pathway appears to occur predominantly during S phase as part of semi-conservative telomere replication. This pathway is preceded by a long-range homology search involving the RAD51-ssDNA presynaptic filament moving micron length distances and the ultimate congregation of telomeres into large clusters where recombination presumably takes place. This movement was shown to be dependent on the Hop2-Mnd1 heterodimer, factors normally associated with homologous chromosome synapsis during meiosis [35]. The requirement for these factors in directing homology search at telomeres is certainly of interest and it is tempting to speculate that GC-rich regions have unique requirements for homology search-and-capture kinetics due to their propensity to form secondary structures, but to date, this remains an unanswered question.

Given these findings, it is reasonable to postulate that the inhibition of telomere maintenance via ALT is ultimately dependent on concomitant inhibition of both amalgamated pathways. Consistent with this, recent work from the O’Sullivan group identified that the vertebrate specific RAD51-interacting protein, RAD51AP1, was required for both RAD51-dependent HR and RAD52-Polδ-mediated telomere extension in ALT cells. This rare implication in both major pathways likely accounts for the exquisite dependence of ALT cancer cells on RAD51AP1 for telomere length maintenance [36].

## 4. ALT-Associated Nuclear Bodies—Sites of Telomere Recombination?

As alluded to previously, it appears that telomeres in ALT cells undergo extensive clustering, which presumably allows the templates for repair to be brought into vicinity for recombination. A further layer of spatial regulation comes from the unique characteristic of ALT cells to encapsulate their telomeres into intranuclear bodies known as ALT-associated PML bodies, or APBs [37]. APBs are generally considered a variant of promyelocytic leukemia nuclear bodies (PML-NBs). PML-NBs, also known as nuclear domain-10 (ND10), Kremer (Kr) bodies or PML oncogenic domains (PODs), are nuclear structures comprising PML and SP100 proteins. These proteins form distinct sub-compartments within the nucleus that are believed to be mediated through multivalent SUMO–SIM (SUMO-interacting motif) interactions which ultimately leads to liquid–liquid phase separation (LLPS) [38,39]. These sub-compartments, which number from 5 to 30, depending on the cell-cycle phase, consist of a 50–100 nm hollow sphere of PML and SP100 proteins [40]. To date, a wide array of proteins have been shown to interact with PML bodies in at least a transient manner [41]. Therefore, PML bodies are considered to be involved in a wide array of nuclear processes.

In normal cells, PML bodies do not contain nucleic acids, with the exception of some evidence of RNA species on the periphery [42]. In ALT cells, however, multiple telomeres cluster together within large PML bodies [43]. Additionally, work to characterize APBs has found a number of DNA repair factors congregate in APBs in ALT. These include RAD51, RAD52, the endonuclease MUS81, replication protein A (RPA), BRCA1 and the MRN complex consisting of MRE11, NBS1, RAD50 and BLM [44,45,46,47]. As such, APBs constitute a complete centre for repair, containing the substrate, template and machinery required for repair.

The cellular mechanisms leading to the assembly of such intricate cellular structures are of profound interest, and, indeed, recent work has shed light on the assembly of PML bodies and the recruitment of telomeric sequence to APBs. In a work by Min et al., fusion proteins containing multiple SUMO and SIM motifs were tethered to telomeres in non-ALT cells. The subsequent assembly of poly-SUMO/SIM-induced condensates through LLPS lead to the assembly of pseudo-PML bodies containing telomeric sequence, akin to a classical APB. Strikingly, however, telomere synthesis was absent from these structures, and only when the Bloom syndrome helicase (BLM) was overexpressed in these cells did multiple ALT markers occur. These features, including heterogeneous telomere length and telomere synthesis, therefore clearly rely on BLM and its activity at APBs [39]. This is in agreement with previous literature suggesting that BLM is required for telomere synthesis in ALT [47].

Furthermore, work has shown that APB formation is linked to the damage of telomeres or replication stress when replicating through telomeric DNA. Indeed, loss of proteins known to suppress replication stress, including FANCM, FANCD2, and SMARCAL1, has been shown to increase APB formation [48,49,50,51]. Additionally, it has been suggested that replication stress at telomeres may lead to MMS21-mediated SUMOylation of Shelterin and other telomere-binding proteins, which, as discussed, may have a role to play in the LLPS process of PML body formation [52]. Finally, recent work has suggested that SUMOylation of TRF2 mediates a feed-forward loop of BIR-driven ALT perpetuation. This would suggest that SUMOylation of TRF2 is an essential step in ALT activation, and, indeed, loss of the SUMO E3-ligase protein PIAS4 leads to reduced recruitment of repair factors to telomeres and, by extension, an ablation of the ALT phenotype [53].

## 5. Replicative Stress at Telomeres—A Problem on Repeat

The implication that ALT is an aberrant form of BIR or MiDAS strongly suggests that a key event in the initiation of ALT is a problem with DNA replication across telomeres. In support of this notion, recent work by the Doksani group showed, by electron microscopy of purified telomeres, that ALT cell telomeres are enriched with single-stranded gaps, often a consequence of replication stress. Furthermore, the spontaneous annealing of these gaps were found to facilitate the formation of so-called internal loops, or i-Loops, which may be further processed to form the extrachromosomal telomeric DNA circles, namely t-circles and c-circles, that have become an indelible marker of ALT cancers [54].

Moreover, several proteins known to modulate DNA replication have been implicated in either repressing or potentiating the ALT pathway. One such protein is SMARCAL1, an ATP-dependent DNA helicase that is proposed to counter replication stress through the promotion of replication fork reversal at stalled forks [55]. Indeed, SMARCAL1 has been shown to localize to ALT telomeres and suppress the ALT phenotype, with loss of SMARCAL1 in ALT increasing telomere clustering events, telomere damage and telomere heterogeneity [49,56]. Furthermore, two ALT-positive cell lines, namely NY and CAL78, have recently been shown to harbor mutations in or display loss of expression of the *SMARCAL1* gene, suggesting that, in certain genomics contexts, the loss of SMARCAL1 may be sufficient to trigger the pathway [57,58]. Akin to SMARCAL1, loss of the fork protection complex (FPC), composed of TIMELESS and TIPIN, leads to an exacerbation in the canonical markers of the ALT pathway in ALT cancer cells [30]. Given the crucial role of the FPC in replication fork progression and maintenance [59], this reinforces the important contribution of replicative stress in the ALT pathway.

As previously discussed, telomeres have been identified to behave as CFSs [11], a trait likely largely attributable to the inherent propensity for telomeres to adopt non-canonical secondary structures, including the G-quadruplex (G4) confirmation and R-loops. Importantly, recent work has shown that telomeres of ALT-positive cancer cells are characteristically enriched in both G4 structures and R-loops and form a linked structure known as a G-loop, where a G4 and an R-loop form on opposing strands [60].

Mounting evidence suggests that both R-loops and G4 structures are an important molecular trigger for the ALT pathway. Treatment with G4-stabilizing ligands, such as RHPS4, has been shown to potentiate c-circles in ALT cells, suggesting that G4 structures can potentiate the ALT pathway [60,61]. Telomeric R-loops are formed through the RAD51- and BRCA2-dependent association of the long non-coding telomeric RNA TERRA within telomeric DNA [62]. Consistent with an important role for R-loops in facilitating ALT, overexpression of RNase H1 in ALT cells to degrade R-loops has been shown to abrogate ALT markers, whereas the loss of RNase H1 or FANCM (a factor also linked to the prevention of R-loop formation), has the opposite effect [48,51,63,64]. As such, it is reasonable to assume that cellular events that lead to the accumulation of G-loops at telomeres may increase the risk to activate the ALT pathway. Indeed, ALT telomeres have been reported to have a reduced compaction of telomeric chromatin and increased levels of telomeric long noncoding RNA “TERRA” expression [63,65]. Interestingly, recent work has shown that reactive oxygen species (ROS)-induced DNA damage at telomeres triggers R-loop accumulation in ALT cells, and this in turn facilitates RAD52 recruitment and POLD3 recruitment, hence facilitating BIR [66]. This raises the interesting possibility that the accumulation of ROS may be pivotal in initiating ALT in certain cancers. In support of this hypothesis, telomerase suppression in a mouse lymphoma model has been found to lead to the development of ALT-positive tumors, and, of note, the resultant tumours characteristically exhibited mitochondrial dysfunction and increased levels of ROS [67].

## 6. ATRX—A Key Player in Suppressing ALT

One common and striking feature of ALT cancers is the remarkable prevalence of mutations in the ATRX–DAXX–H3.3 axis. While mutations in ATRX are generally rare in cancers overall, with less than 1% of 84,000 cases in The Cancer Genome Atlas containing an ATRX mutation, rates of ATRX mutations in ALT cancers are comparatively much higher. When assessing panels of ALT cancers, work has previously shown that over 90% of ALT cancers display undetectable levels of ATRX protein and mutations or deletions in the *ATRX* gene [68]. These observations strongly suggest that ATRX has a key role in suppressing tumorigenesis via ALT. Indeed, previous work by our lab and others has demonstrated that ectopic expression of ATRX leads to a DAXX-dependent suppression of the ALT pathway [69,70].

ATRX is a chromatin remodeling factor of the Snf2 family (reviewed in Reference [71]) that, together with the histone chaperone DAXX, facilitates the incorporation of the histone variant H3.3 into defined genomic sites, including telomeric chromatin, in a replication independent chromatin assembly pathway [72,73,74]. In the last decade, a number of studies have implicated ATRX in a plethora of roles related to the maintenance of genome stability that likely provide mechanistic insight into its role as a suppressor of the ALT pathway. One major role of ATRX appears to be the regulation of non-canonical DNA secondary structures during DNA replication. Recent work has shown that ATRX associates with multiple subunits of the MCM-replication complex subunits, with loss of ATRX leading to G4-structure accumulation at newly synthesized DNA [75,76] and increases in the level of R-loops [77]. ATRX has multiple reported roles in DNA replication which may or may not be linked to this activity, including the prevention of replication fork stalling, potentiation of fork restart and the prevention of excessive nucleolytic degradation of stalled forks [78,79,80,81]. A recent analysis of proteins recruited to CFSs identified ATRX as a pivotal player in CFS stability upon induction of replicative stress, with ATRX being recruited to a subset of CFSs with DAXX in an FANCD2-dependent manner [82]. The notion of functional interactions between the ATRX/DAXX complex and FANCD2 is further evidenced in recent work showing that ATRX and FANCD2 localize to stalled forks and recruit the resection factor CtIP and promote MRE11 exonuclease-dependent fork restart. Furthermore, this mechanism was dependent on the deposition of H3.3 through ATRX/DAXX, demonstrating that coordinated deposition is a critical event in re-initiation of replicative DNA synthesis [83].

Additionally, evidence exists for a role of ATRX in facilitating DNA double-strand break (DSB) repair, with roles reported in both the major DSB repair pathways, HR [83,84] and NHEJ [85]. Which of these roles are required for ATRX to prevent either the induction or maintenance of the ALT pathway is, to date, unclear.

Given the prevalence of ATRX mutations in ALT, and its ability to suppress the ALT phenotype, one might expect that ATRX loss is uniquely responsible for ALT development. However, previous work has shown that, in fact, loss of ATRX is generally insufficient to trigger ALT in most cells [68,70,81,86], with the notable exception of a minority of glioma cell lines [87]. Given that this induction was specific to these cell lines, it is likely that additional genetic or epigenetic events occur alongside ATRX loss to facilitate induction of ALT in human cancer. Consistent with this, point mutation of Isocitrate Dehydrogenase 1 (IDH1) (R132H), which is frequently found to be coincident with ATRX mutations in certain cancers, including glioma [88], is sufficient to create tumorigenic cells with ALT characteristics [89]. R132H IDH1 mutations occur early in the development of glioma and display neomorphic activity that converts α-ketogluterate to 2-hydroxyglutarate (2-HG) [90]. Accumulation of 2-HG in turn results in the inhibition of a plethora of enzymes with known functions in both DNA and histone methylation and subsequent alterations in gene expression [91,92,93]. Mukherjee and colleagues further demonstrated that the induction of ALT characteristics was attributable to the downregulation of specific genes: the Shelterin component RAP1, the DNA-repair scaffolding protein XRCC1 and DNA Ligase 3, which cooperate in DNA single-strand break repair (SSBR) and MMEJ [94,95,96]. As such, it appears that the cooperative occurrence of telomere dysfunction or telomeric replicative stress, such as that arising from Shelterin disruption [11], and ATRX loss is a potential route to ALT activation. The reduction in levels of XRCC1 suggests that altering the balance between the repair pathways available at telomeres may result in a favoring of HR repair and ALT. MMEJ has previously also been linked to the fusion of some forms of dysfunctional telomeres which would likely be prohibitive to both cellular survival and engagement of BIR [97]. As such, prevention of these fusion events may serve as an additional requisite for ALT. More recent work has suggested that it is the IDH1 R132H-mediated inhibition of KDM4B activity that is responsible for ALT induction in the context of ATRX loss in a mouse embryonic-stem-cell model, suggesting that there may be different dependencies for different cellular contexts, while underlining the likely importance of changes in chromatin composition in ALT activation [98].

The aforementioned roles of ATRX in the response to replication stress and DNA repair could explain the involvement of ATRX in ALT, wherein DNA double-strand breaks (DSBs) at telomeres are required for the initiation of BIR (see Figure 1 for synopsis). However, until recently it was unclear as to how telomeric DSBs repaired by aberrant BIR would lead to telomere lengthening. Generally speaking, homology driven DSB repair processes are carefully orchestrated to ensure that strand invasion mediated by RAD51 (or RAD52 in the case of BIR) remains in-register (sister strands are aligned to allow for faithful strand invasion). Recent work, however, has highlighted an additional role for ATRX in the maintenance of telomere cohesion. Specifically, that loss of ATRX results in a telomere-specific loss of cohesion of sister chromatids [99]. This observation is significant, in that it may explain why out-of-register BIR occurs so frequently in ALT cancers, but not in non-ALT cells (Figure 1).

## 7. Closing Thoughts

Exploring the mechanisms underpinning the ALT pathway has extensively highlighted the imperative importance of careful coordination and regulation of the cell’s response to DNA double-strand breaks at many levels, including chromatin compaction, DSB mobility and the faithful clustering and then alignment of broken chromatids to allow for faithful repair. The work explored here has highlighted how multi-faceted repair of certain double-strand breaks can be, with multiple converging alternative pathways. This is important when looking to identify new drug targets for the treatment of ALT cancers. Understanding the pathways and players involved in ALT will hopefully facilitate the rapid development of targeted therapies in the not-too-distant future.

## Figures and Tables

**Figure 1 genes-12-01734-f001:**
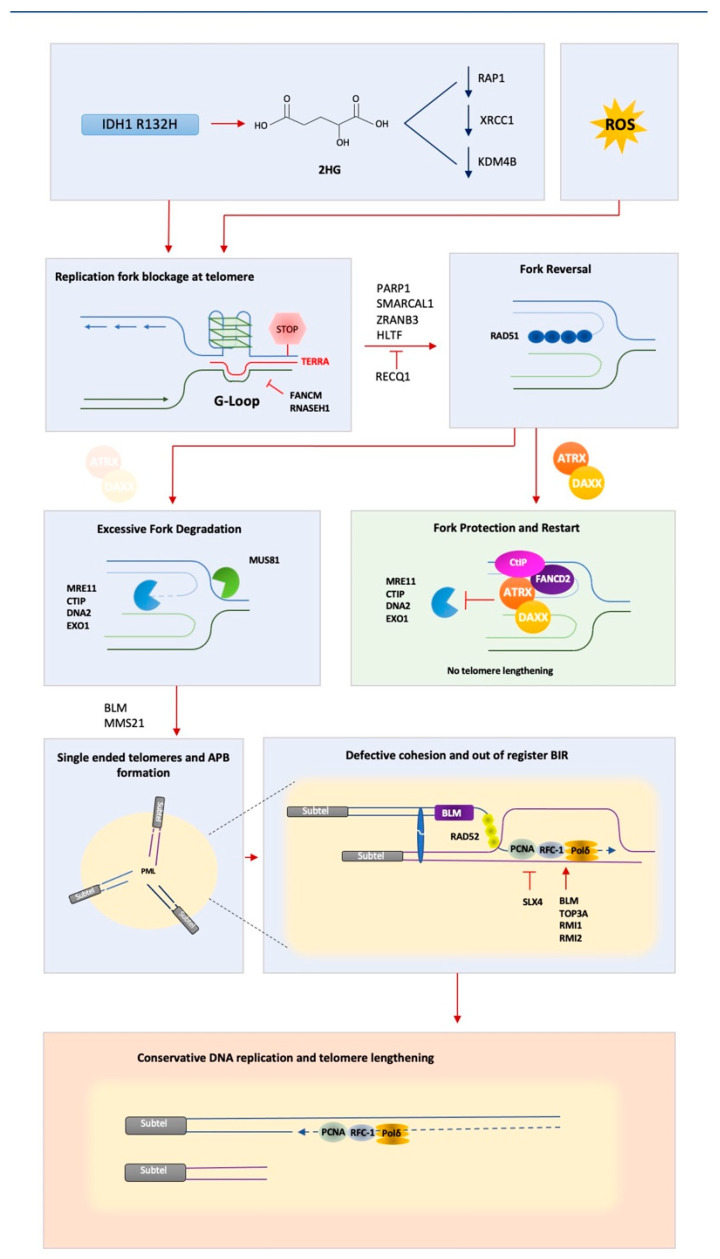
Proposed model for activation of the ALT pathway. Mutation of IDH1 R132H leads to accumulation of 2-hydroxyglutarate, which leads to inhibition of multiple enzymes with functions in both DNA and histone methylation. This has been linked to decreases in expression of XRCC1 and RAP1. Loss of XRCC1 prevents formation of telomeric fusions in ALT cancer cells, whereas loss of RAP1 disrupts Shelterin, potentially leading to the formation of DNA secondary structures on telomeric DNA such as G-loops. The accumulation of reactive oxygen species has also been linked to ALT cancers and the formation of telomeric R-loops. Upon stalling at a replication barrier, replication forks are reversed/regressed. Fork regression is mediated by RAD51 and other DNA translocases, including SMARCAL1, ZRANB3 and HTLF. In the presence of ATRX/DAXX, forks are stabilized and restarted through interaction of ATRX with FANCD2 and CTIP. In the absence of ATRX, reversed forks undergo excessive MRE11-dependent degradation and cleavage by the MUS81 nucleases, leading to the formation of one-ended telomeric DSBs. Damage-induced MMS21-mediated SUMOylation of telomere proteins results in BLM-dependent PML recruitment to telomeres and triggers APB formation through SUMO/SIM-mediated LLPS. PCNA-RFC-Polδ-mediated BIR then occurs between clustered telomeres within the APB in a process that is promoted by BTR complex (BLM, TOP3A, RMI1 and RMI2) and inhibited by SLX4. Strand invasion is out of register, owing to ATRX loss, leading to telomere lengthening through conservative replication.

## Data Availability

Not applicable.

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
