# Peer review of "Alternative Lengthening of Telomeres: Lessons to Be Learned from Telomeric DNA Double-Strand Break Repair"

_genes, 2021, doi:10.3390/genes12111734_

Round 1
Reviewer 1 Report
In this manuscript, Kent and Clynes review the work carried over decades by scientists studying the ALT pathway. The authors have done a fair review of the dense work on factors involved in the Telomere maintenance mechanism used in 10-15% of cancer cells to avoid cell death and senescence. The authors have also nicely tried to move downstream to the earliest driver that can initiate this phenomenon but keeping in mind that it is likely a multi factorial induction. The review is well balance on all aspects and don’t have any major modifications to request. I am happy with this version of manuscript providing the 2 following minor modifications:
- Sentence from 26 to 30 is too long and unclear.
- Paragraph 164-171: it is important to cite BLM factor recruited to PML here:
Ref: D.J. Stavropoulos, P.S. Bradshaw, X. Li, I. Pasic, K. Truong, M. Ikura, M. Ungrin, M.S. Meyn The Bloom syndrome helicase BLM interacts with TRF2 in ALT cells and promotes telomeric DNA synthesis Hum. Mol. Genet., 11 (2002), pp. 3135-3144
Author Response
In this manuscript, Kent and Clynes review the work carried over decades by scientists studying the ALT pathway. The authors have done a fair review of the dense work on factors involved in the Telomere maintenance mechanism used in 10-15% of cancer cells to avoid cell death and senescence. The authors have also nicely tried to move downstream to the earliest driver that can initiate this phenomenon but keeping in mind that it is likely a multi factorial induction. The review is well balance on all aspects and don’t have any major modifications to request. I am happy with this version of manuscript providing the 2 following minor modifications:
Thank you for your kind comments, we are pleased you enjoyed our review!
- Sentence from 26 to 30 is too long and unclear.
Thank you, we have simplified this sentence as part of our newly written abstract (see lines 17 to 21)
- Paragraph 164-171: it is important to cite BLM factor recruited to PML here:
Ref: D.J. Stavropoulos, P.S. Bradshaw, X. Li, I. Pasic, K. Truong, M. Ikura, M. Ungrin, M.S. Meyn The Bloom syndrome helicase BLM interacts with TRF2 in ALT cells and promotes telomeric DNA synthesis Hum. Mol. Genet., 11 (2002), pp. 3135-3144
We have now added BLM to the list of factors recruited to PML and added this reference (see line 261).
Reviewer 2 Report
This manuscript reviews what is known about the maintenance of telomere length in ALT cancer cells. The authors describe how the ALT pathway is triggered after replication fork blockage at telomeres due to secondary structures, and the role of ATRX in preventing activation of the pathway, as well as the mechanism leading to telomere lengthening.
This review is well-written and easy to read. It provides a concise overview of the topic, and the final figure helps to summarise the findings presented in the manuscript. I have some minor comments listed below.
- The abstract is a little big vague. The presentation of the different DSBR pathways would fit better in the main text as introduction, and the abstract should focus on the main messages of the review.
- Figure 1: Many proteins appearing in the figure are not described in the text nor in the figure legend. It would be good to at least mention them.
- Line 89: telomerase reverse transcriptase is TERT, not TERC
- Lines 107-112: Please clarify how the unified pathway can lead to both type I and type II survivors.
- Line 118 and 304: HR instead of HDR
- Line 139: Please specify why “it is tempting to speculate that GC rich regions have unique requirements for homology search and capture kinetics”
- Line 196: MiDAS
- Lines 203 and 227: either c-circle or C-circle.
- Line 236: one word is missing “a reduced compaction of telomeric … and”
- Lines 293 and 335: 2-HG is 2-hydroxyglutarate
- Line 302: please reformulate the sentence “the downregulation of XRCC1 loss…”
- Line 337: RAP1 instead of RAPH1?
- I am not familiar with the expression “out of register”, which is used throughout the manuscript, maybe reformulate in a clearer way.
Author Response
This manuscript reviews what is known about the maintenance of telomere length in ALT cancer cells. The authors describe how the ALT pathway is triggered after replication fork blockage at telomeres due to secondary structures, and the role of ATRX in preventing activation of the pathway, as well as the mechanism leading to telomere lengthening.
This review is well-written and easy to read. It provides a concise overview of the topic, and the final figure helps to summarise the findings presented in the manuscript. I have some minor comments listed below.
Thank you for your kind comments, we are pleased you enjoyed our review and very much appreciate the time you have taken!
- The abstract is a little big vague. The presentation of the different DSBR pathways would fit better in the main text as introduction, and the abstract should focus on the main messages of the review.
Thank you for the suggestion. We have now written a more focused abstract and as suggested moved the discussion on DSBR pathways into the main text.
- Figure 1: Many proteins appearing in the figure are not described in the text nor in the figure legend. It would be good to at least mention them.
We have now included a description of the role of the BTR and SMX complexes in the main text (line 191 – 195) and included them in the figure legend.
- Line 89: telomerase reverse transcriptase is TERT, not TERC
Thank you for spotting this mistake, it has now been corrected (line 153).
- Lines 107-112: Please clarify how the unified pathway can lead to both type I and type II survivors.
We have added some further explanation of these findings (line 175 – 179).
- Line 118 and 304: HR instead of HDR
This has now been corrected (lines 185 and 496)
- Line 139: Please specify why “it is tempting to speculate that GC rich regions have unique requirements for homology search and capture kinetics”
We speculate that this may be due to a propensity to form secondary structures, although there is no data to support this so just conjecture at this stage. This has been added to the text (line 251)
- Line 196: MiDAS
Corrected (Line 325)
- Lines 203 and 227: either c-circle or C-circle.
Corrected to c-circle (line 356)
- Line 236: one word is missing “a reduced compaction of telomeric … and”
‘chromatin’ added to line 365
- Lines 293 and 335: 2-HG is 2-hydroxyglutarate
Corrected (line 485 and 539)
- Line 302: please reformulate the sentence “the downregulation of XRCC1 loss…”
Corrected (line 494)
- Line 337: RAP1 instead of RAPH1?
Corrected (Line 542)
- I am not familiar with the expression “out of register”, which is used throughout the manuscript, maybe reformulate in a clearer way.
We have added a clarification on line 459-46.
Reviewer 3 Report
The authors have written a well-organized and thoughtful review on the ALT pathway and its role in cancer. The article is suitable for publication pending a few minor corrections:
- On lines 168 and 169, BRCA1 is defined, but it is mentioned previously without definition in the abstract.
- On lines 269, 279, and 280, the authors define terms that have been previously defined in the manuscript.
- The panels and text in the figure are too small. Use as much of the available space on the page as possible and if the legend spills over a page it is no big deal.
Author Response
The authors have written a well-organized and thoughtful review on the ALT pathway and its role in cancer. The article is suitable for publication pending a few minor corrections:
Thank you for your kind words and time, we are glad you enjoyed our review!
- On lines 168 and 169, BRCA1 is defined, but it is mentioned previously without definition in the abstract.
This has been corrected, we now define BRCA1 and BRCA2 on line 31 but not later in the manuscript.
- On lines 269, 279, and 280, the authors define terms that have been previously defined in the manuscript.
We have altered this (lines 416, 426 and 427)
- The panels and text in the figure are too small. Use as much of the available space on the page as possible and if the legend spills over a page it is no big deal.
We have made the figure bigger.